# Nurses’ Perceptions of the Quality of Procedural Sedation in Children Comparing Different Pharmacological Regimens

**DOI:** 10.3390/children9071068

**Published:** 2022-07-18

**Authors:** Antonietta Curatola, Martina D’Agostin, Elena Favaretto, Giada Vittori, Viviana Vidonis, Tamara Strajn, Nicole De Vita, Alessia Saccari, Egidio Barbi, Luisa Cortellazzo Wiel

**Affiliations:** 1Department of Pediatrics, Università Cattolica del Sacro Cuore, 00168 Rome, Italy; antoniettacuratola01@gmail.com; 2Department of Pediatrics, University of Trieste, 34127 Trieste, Italy; elenafavaretto93@gmail.com (E.F.); egidio.barbi@burlo.trieste.it (E.B.); 3Institute of Maternal and Child Health IRCCS Burlo Garofolo, 34137 Trieste, Italy; giada.vittori@burlo.trieste.it (G.V.); viviana.vidonis@burlo.trieste.it (V.V.); tamara.strajn@burlo.trieste.it (T.S.); nicoledevita@burlo.trieste.it (N.D.V.); alessia.saccari@burlo.trieste.it (A.S.); luisacortellazzowiel@gmail.com (L.C.W.); 4Bradford Teaching Hospitals NHS Foundation Trust, Bradford BD9 6RJ, UK

**Keywords:** pediatric procedural sedation, nurses’ satisfaction, propofol, ketamine, dexmedetomidine, midazolam

## Abstract

Nurses play a pivotal role during pediatric procedural sedation and their perspective is an important indicator for the quality of care. The aim of this study is to examine nurses’ satisfaction comparing four different pharmacological regimens used for pediatric sedation outside of the operating room. A prospective observational study was conducted in a third-level pediatric teaching hospital, involving all the nurses with experience in the field of pediatric procedural sedation. A 13-item survey was used to assess the level of nurses’ satisfaction for the quality of sedation with four different analgesic–sedative drugs. Fifty-one questionnaires were completed by pediatric nurses, with a median length of experience of 10 years. Regarding the overall quality of the sedation, the highest median satisfaction scores were observed for propofol (8, IQR 7–9), dexmedetomidine (8, IQR 6–8) and midazolam (8, IQR 7–9). Ketamine (5, IQR 3–7) displayed the lowest score. When asked to rate their level of perceived safety, nurses gave high scores to all the four drugs studied, with no statistically significant difference between them. Non-pharmacological techniques during procedural sedation were judged as important by 38 (74.5%) nurses. According to this sample of pediatric nurses, the best quality of procedural sedation in children outside of the operating room is obtained with propofol, dexmedetomidine and midazolam. During procedural sedation, nurses feel safe overall, regardless of the pharmacological regimen used. Moreover, they highlighted the relevance on non-pharmacological approaches in the preparation of the child for the procedure.

## 1. Introduction

Diagnostic and therapeutic procedures can be stressful for children, and their perception of pain can be enhanced by emotional factors, such as anxiety, distress, or anger [1]. The best management relies on careful patient assessment and effective use of non-pharmacologic and pharmacologic treatments [2]. Indeed, pediatric sedation represents the standard of care to reduce the fear and anxiety of the patients and of their parents, while optimizing the length of the procedures. Procedural sedation is defined by the American College of Emergency Physicians as “a technique of administering sedatives or dissociative agents with or without analgesics to induce a state that allows the patient to tolerate unpleasant procedures while maintaining cardiorespiratory function” [3].

Commonly used pharmacologic agents for pediatric procedural sedation are propofol, dexmedetomidine, ketamine, fentanyl and midazolam [4]. As a matter of fact, any of these agents or various possible combinations can be relied upon by sedationists to provide the best care to each child. Establishing the best sedative choice for each patient is of relevance, especially for children with chronic illnesses who will likely undergo repeated procedures throughout their life.

In the pediatric field, parents’ satisfaction represents a quality indicator in healthcare. Furthermore, awareness of parents’ perception about the different available drug regimens may be relevant to guide the pediatrician in tailoring the best pharmacological option to each child. In the literature, caregivers’ satisfaction for pediatric sedation is rated as high, regardless of the pharmacological regimen used [5,6,7]. However, caregivers’ overall perception is likely influenced by several variables (from parking facilities to waiting times, staff pleasantness, specific procedure’s related issues), and may be also limited by the fact that the sedation regimen examined is often the only one they ever experienced, with no or limited experience of the other existing options. On the other hand, pediatric nurses may be more objective in judging the sedation quality for their patients. In fact, nurses play a pivotal role during procedural sedation, since they are involved in both patients’ and parents’ preparation, drug administration and recovery monitoring. We believe that understanding nurses’ opinion could help in improving the standard of care. Remarkably, very little is known about pediatric nurses’ perception of the quality of sedation performed with the different available pharmacological regimens [7].

The aim of this study was to examine nurses’ perception of the analgesic–sedative drug regimens used in the pediatric age outside of the operating room.

## 2. Materials and Methods

### 2.1. Survey Structure

A cross-sectional study was conducted at the University teaching, tertiary children’s hospital, Institute for Maternal and Child Health IRCCS Burlo Garofolo of Trieste, Italy, involving all the nurses with experience in the field of pediatric peri-procedural sedation.

As no previous validated instrument existed, a questionnaire was specifically developed for the aim of the study by a group of pediatricians, pediatric anesthetists and pediatric nurses, coordinated by a professional with specific experience in the design of self-report surveys. Before developing the final version, the questionnaire was tested on a convenience sample of 30 nurses, assessing the accuracy of the questions and their applicability on a larger scale. Subsequently, the questions were refined or modified based on the recommendations obtained from the initial participants. Then the survey was administered twice to the nurses over a period of six months. A test–retest reliability showed good reliability on the results obtained in the two different point in time (Pearson Correlation Coefficient = 0.81).

An email containing a link to an online survey was sent to the institutional addresses of the eligible nurses. Participants completed an anonymous online survey. The survey consisted of thirteen multiple-choice questions (Appendix A), regarding the nurses’ experience, the preferred route of administration of sedative drugs, the perceived quality of the overall sedation and safety with each pharmacological regimen according to a 10-point numerical scale (NRS), the frequency of adverse events (AE) and the need for a rescue drug. For some questions more than one answer was possible. Finally, the survey contained two open-narrative questions to provide other comments or feedback about the perception of the different pharmacological options. Participants were allowed to answer all or only some of the questions according to their own experience. The surveys were collected through a Google web-based platform. The study was approved by the appropriate Institutional Review Board RC 34/18.

### 2.2. Sedation Standard-of-Care in Our Hospital

As per institutional protocol, in this facility pediatricians and pediatric residents are specifically trained in sedation by joining the Pediatric Sedation Unit and performing daily sedations for both painful and non-painful procedures outside the operating room. This Unit is supervised by two pediatric anesthetists who guarantee training, monitoring and emergency response. Trained pediatric nurses aid during admission, intravenous (IV) line establishment, monitoring of sedation and recovery. Pediatric nurses receive specific training concerning the main principles of sedative drug pharmacology and AE, relevance of airways patency, patients’, drugs’ and specific procedures’ sedation related risk factors and contraindications, use of non-pharmacological techniques, use of topic anesthesia, monitoring skills (including pulse oximetry and capnography), oxygen administration (bag valve mask ventilation, pharyngeal airway, nasal trumpet), recovery and discharge monitoring standards. Regular audits are performed to discuss debated issues and clinical cases.

The most used analgesic–sedative pharmacological regimens are: propofol, ketamine, midazolam and dexmedetomidine, alone or in association. The pharmacological strategy is chosen in view of several factors, including the patient’s age, history, physical examination, procedure performed, degree of expected pain, history of allergy or previous adverse drug reactions and expected difficulty in establishing an IV line according to the child’s previous history or Difficult Intravenous Access (DIVA) score [8].

Dealing with children requiring sedation for non-painful procedures involving a motionless status such as upper and lower endoscopies, MRI/CT scans or scintigraphy and painful procedures such as bone marrow aspirates/biopsies, lumbar punctures, or fracture reductions, moderate to deep sedation is usually required.

Generally, dexmedetomidine is used for long imaging procedures (e.g., MRI, scintigraphy), propofol for non-painful or minimally painful procedures requiring deeper levels of sedation such as endoscopies, lumbar punctures or bone marrow aspirates, and ketamine, either alone or in combination with propofol, for painful procedures such as bone, liver and kidney biopsies, fracture reductions or teeth extractions. Premedication with oral (PO) midazolam or intranasal (IN) dexmedetomidine is generally used to facilitate venous cannulation in young children (under six years of age) or children with high DIVA score; IV midazolam before IV propofol is used for older children experiencing fear and anxiety before starting propofol infusion. Eutectic Mixture of Local Anesthetics cream is usually applied locally any time a puncture is performed. Distraction techniques, such as playing with favorite toys, watching videos/movies, reading a book or listening to music/songs are routinely used with children of all ages in addition to the specific pharmacological regimen. A standardized consent for sedation is required by parents, who are actively involved in supporting their child during venous cannulation or drug administration, and as a rule, remain next to their child until an adequate sedation level is achieved.

### 2.3. Statistical Analysis

All statistical analyses were conducted using R software, version 4.1.0 (R Foundation for Statistical Computing, Vienna, Austria). All descriptive statistics data are presented as number and percentage (%) or median and interquartile range (IQR) as appropriate. Kruskal Wallis rank sum test with Bonferroni post-test adjustment were used to compare categorical or continuous variables between the different groups. A *p*-value less than 0.05 was considered as statistically significant.

## 3. Results

The survey was sent to 60 nurses. Nine nurses did not participate in the survey. Thus, in total, 51 questionnaires were collected. The participating nurses were employed in the following departments: 18 in the Pediatric Clinic, 13 in the Pediatric Day Hospital, 13 in the Oncology Department, 5 in the Orthopedics Department and 2 in the radiology department. The nurses’ median years of experience were 10 years (IQR 2.75–23.75). Further, 27 nurses (53%) attended less than 10 peri-procedural sedations per month, 14 (27.4%) from 10 to 20, 8 (15.7%) from 21 to 30 and 2 (3.9%) more than 30. Propofol was indicated as the favorite sedative regimen by 25 nurses (49%). Further, 15 nurses (29.5%) preferred midazolam alone, 9 (17.6%) dexmedetomidine and 2 (3.9%) ketamine. When asked about their favorite routes of administration of sedative drugs, 28 (55%) nurses indicated the IV route, 15 (29.5%) the PO and IN route, 7 (13.5%) the IN route and 1 (2%) the intramuscular (IM) route. Distraction techniques during procedural sedation were judged as very important by 31 (60.8%) nurses, important by 7 (13.7%) nurses, scarcely important by 9 (17.6%) nurses and not important at all by 4 (7.9%) nurses.

Forty (78.4%) nurses wished to have venous access established in all children while 11 (21.6%) wished it was placed at least in the most difficult cases (e.g., children with genetic conditions, obesity, current respiratory illness or high DIVA score). 

### 3.1. Nurses’ Satisfaction with the Overall Quality of Sedation

Regarding the overall quality of the sedation, the highest median satisfaction scores from the patients’ perspective were observed for propofol, dexmedetomidine and midazolam. Ketamine displayed the lowest score. The median satisfaction scores for all the pharmacological regimens are shown in Table 1. Overall, a statistically significant difference was observed between the medians of the nurses’ satisfaction scores for the different pharmacological regimens (Kruskal–Wallis *p*-value 0.005): ketamine displayed significantly lower satisfaction scores than propofol (*p*-value Bonferroni < 0.001), midazolam (*p*-value Bonferroni 0.005) and dexmedetomidine (*p*-value Bonferroni 0.003), whereas propofol, dexmedetomidine and midazolam showed the same satisfaction scores.

To better understand what drove nurses’ satisfaction, they were asked to indicate the aspects of the sedation process that influenced their satisfaction more. Almost all the nurses (90%) reported that their satisfaction was negatively influenced by both the presence of adverse events and parental/children unsatisfaction. The quality and timing of recovery, the type of sedative drug, the route of administration, the use of rescue drugs and the nursing staff availability were less reported in descending order, as factors affecting nurses’ satisfaction.

### 3.2. Nurses’ Perception of Safety and Adverse Effects

Nurses’ opinion about safety is reported in Table 1. No statistically significant difference was observed between the drugs used (Kruskal–Wallis *p*-value 0.08). Nurses gave high scores to almost all the drugs used. Secondly, nurses were asked to indicate all the AE they had experienced with each drug and to report how much the presence of an AE during and after the sedation reduced their overall satisfaction (Table 2 and Table 3). Furthermore, the need for a rescue medication (mainly antiemetic drugs for vomiting, flumazenil for midazolam-related paradoxical reaction or dextrose infusion for low-blood-sugar levels in patients with long awakening times) was reported by 40 (78.4%) nurses after sedation with ketamine, 16 (31.4%) with midazolam, and 5 (9.8%) with propofol. Finally, when asked for additional comments to improve the standard of care in the sedation unit, most of the nurses suggested to carry out training courses and updates on procedural sedation more frequently, to improve communication between clinicians and nurses and to better implement the non-pharmacological and distraction techniques.

## 4. Discussions

Nurses’ perspective is an important indicator for the quality of care in pediatric sedation practice. This is an interesting and understudied area of work. In the literature, few studies are available about caregivers’ perception of pediatric peri-procedural sedation and in most of the cases, parents’ satisfaction is high regardless of the sedative drug used [5,8,9]. While several factors are reported to influence parental satisfaction, such as environment, timeliness and communication issues, the most relevant one is the quality of the child’s recovery from sedation [5,6,9,10]. However, parents’ perceptions may be limited by a lack of knowledge about the different available drug regimens and by the experience of only one or limited types of sedation in their child. Very few data are available on nurses’ perception of periprocedural sedation. In a study performed to implement guidelines for the management and treatment of pain in the pediatric cardiac intensive unit, team members (physicians and nurses) were surveyed before and after the implementation of guidelines, and improvements in teamwork and patient care were demonstrated [11]. To our knowledge, this is the first study analyzing nurses’ satisfaction for peri-procedural sedations in children comparing different sedative drugs. We believe that nurses’ perception plays a relevant role whenever different choices are available, since teamwork between physicians and nurses can be the key to more successful patient- and family-centered care.

This study shows that in the nurses’ opinion, the best sedation regimens are propofol, dexmedetomidine and midazolam, whereas all nurses were dissatisfied with the use of ketamine. In addition, we found that the nurses’ satisfaction for propofol increased with the number of sedations per month attended by the responding nurse. Regarding the nurses’ preferred route of administration of sedative drugs, the IV route and the IN combined with the PO route were favored in this survey.

We found that 74.5% of the surveyed nurses rated the distraction techniques as important. The latter are simple, cost effective, easy to learn and not time-consuming. Previous studies analyzed non-pharmacological interventions in pediatric patients to reduce the pain associated with minimally painful procedures, such as venous cannulation, and showed that these techniques, alone or in association with pharmacological treatments, can reduce pain and fear, beside promoting the cooperation of the child and their parents [12]. Nurses’ awareness of the relevance of non-pharmacological techniques and adequate communication with patients and parents is a strong reminder for all physicians of the importance of this pivotal component of care.

Regardless of the pharmacological regimen used, nurses reported feeling safe during procedural sedation, even if the latter is not completely risk-free [13]. This result should encourage the systematic use of sedative drugs in adequate settings with defined levels of training and monitoring, for children who undergo diagnostic or therapeutic procedures causing pain or excessive stress. Moreover, this study showed that an available venous access is considered by nurses as a fundamental requirement, especially in children with higher risks of severe complications (children with genetic conditions, obesity, current mild respiratory illness or high difficult intravenous access score).

Throughout this study, the main factors affecting the level of nurses’ satisfaction were the presence of AE and parental/children satisfaction. What we already know about caregivers’ satisfaction for pediatric sedation is that experiencing post-sedation respiratory distress, vomiting, irritability, hyperactivity and hallucinations are all independent risk factors for parental lower satisfaction scores [9]. Similar results were obtained also in this nurses’ survey. In fact, when assessing which AE reduced nurses’ satisfaction more, respiratory distress, hallucinations and nausea/vomiting had the largest impact.

The AEs most frequently reported by nurses are the ones widely known in the literature [4,13]; in particular, sleepiness for propofol, midazolam and dexmedetomidine, nausea, vomiting, irritability and hallucinations for ketamine. As a matter of fact, ketamine was reported in nurses’ perception to have more AEs and need for a rescue medication. This could be the main reason why ketamine displayed significantly lower satisfaction scores compared to the other pharmacological regimes. The low appreciation for ketamine by nurses in this study should prompt some considerations. As a matter of fact, ketamine is considered a gold standard by pediatricians due to its safety and effectiveness [14]. However, in the setting of minimally to moderately painful procedures, propofol may be equally effective and safe.

We are not assuming that nurses’ perceptions are an absolute proxy for quality of sedation. Even if nurses’ view may be highly sympathetic with the children and parents’ perspective, some bias should be considered. For instance, a long recovery time may be perceived by a nurse as a limit more than by a parent or a physician. Moreover, since in nurses’ perception, quality of recovery and number of adverse events played a major role, the evaluation of each drug’s impact may have been biased by a sectorial perspective, not considering other issues, such as safety and feasibility of administration, for each specific child and setting. Overall, these considerations limit the meaningfulness of the specific drug evaluations. However, even if limited, we decided to maintain this view, as it is often reported in real life by the staff. We believe that it should be taken into account in the perspective of a constructive team working, seeking patient- and family-centered care. Overall, this evidence further suggests that nurses’ opinion should be endorsed in the setting of teamwork when defining a sedation strategy for a specific patient.

This study has some limitations. Firstly, a non-validated questionnaire was used as a validated instrument to measure the specific outcome of this study, as this was not available in the literature. Secondly, the number of surveyed nurses was relatively small as the study was limited to our single center. In addition, many nurses may not have enough experience with all types of sedative drugs, so their satisfaction may also have been biased by this limit.

The points of strength are the enrolment of pediatric nurses with relevant experience in peri-procedural pediatric sedation outside the operating room and the use of a questionnaire developed in two steps through a joint effort of physicians, nurses and parents. Finally, it offers an interesting insight into why nurses favor one drug regimen over another. The aspects of the sedation process that seem to influence the nurses’ satisfaction most are the presence of AE and parental/children satisfaction, suggesting that these areas should be targeted to enhance the quality of children’s care.

Drugs will change over the years but understanding the components of the quality of care is an enduring piece of knowledge.

## 5. Conclusions

In conclusion, this study showed how the presence of AE and parental/children satisfaction influence most nurses’ satisfaction of procedural sedation outside the operating room. Moreover, nurses’ opinion highlighted the relevance on non-pharmacological approaches in the preparation of the child for the procedure.

## Figures and Tables

**Table 1 children-09-01068-t001:** Nurses’ perception about overall quality of sedation and safety using four different pharmacological regimens.

	Overall Quality of Sedation		Safety of Sedation	
Pharmacological Regimens	Median	IQ Range	Median	IQ Range
Propofol	8.00	7.00–9.00	8.00	7.00–9.00
Dexmedetomidine	8.00	6.00–8.00	7.50	4.00–8.00
Midazolam	8.00	7.00–9.00	9.00	7.00–9.50
Ketamine	5.00	3.00–7.00	8.00	6.00–9.00

IQ = interquartile.

**Table 2 children-09-01068-t002:** Nurses’ opinion about the frequency of adverse effects experienced with four different pharmacological regimens (*n*, %).

	Sleepiness	Irritability	Restlessness -Hyperactivity	Hallucinations	Unsteadiness	Headache	Dizziness	Alterations in Appetite	Nausea -Vomiting	Respiratory Distress	None	NA
**Propofol**	29 (56.9)	3 (5.9)	0 (0)	2 (3.9)	0 (0)	0 (0)	0 (0)	0 (0)	1 (2)	0 (0)	12 (23.5)	5 (9.8)
**Dexmedetomidine**	16 (31.4)	7 (13.7)	5 (9.8)	2 (3.9)	2 (3.9)	0 (0)	1 (2)	1 (2)	3 (5.9)	0 (0)	7 (13.7)	12 (23.5)
**Midazolam**	19 (37.3)	12 (23.5)	8 (15.7)	4 (7.8)	4 (7.8)	0 (0)	3 (5.9)	1 (2)	3 (5.9)	0 (0)	6 (11.8)	5 (9.8)
**Ketamine**	5 (9.8)	10 (19.6)	13 (25.5)	21 (41.2)	6 (11.8)	0 (0)	2 (3.9)	7 (13.7)	18 (35.3)	0 (0)	2 (3.9)	6 (11.8)

NA = not applicable.

**Table 3 children-09-01068-t003:** The impact of adverse effects in reducing nurses’ satisfaction on procedural sedation (*n*, %).

	Sleepiness	Irritability	Restlessness -Hyperactivity	Hallucinations	Unsteadiness	Headache	Dizziness	Alterations in Appetite	Nausea -Vomiting	Respiratory Distress
**No influence**	29 (56.8)	12 (23.5)	8 (15.7)	9 (17.7)	15 (29.4)	12 (23.5)	9 (17.6)	22 (43.1)	6 (11.8)	6 (11.8)
**Low influence**	19 (37.3)	10 (19.6)	13 (25.5)	8 (15.7)	25 (49.1)	18 (35.4)	16 (31.3)	25 (49.1)	13 (25.6)	3 (5.9)
**Medium influence**	1 (2)	14 (27.5)	16 (31.3)	17 (33.3)	7 (13.7)	16 (31.3)	18 (35.4)	2 (3.9)	16 (31.3)	7 (13.7)
**High influence**	2 (3.9)	15 (29.4)	14 (27.5)	17 (33.3)	4 (7.8)	5 (9.8)	8 (15.7)	2 (3.9)	16 (31.3)	35 (68.6)

## Data Availability

Not applicable.

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
