# Peer review of "Nurses’ Perceptions of the Quality of Procedural Sedation in Children Comparing Different Pharmacological Regimens"

_children, 2022, doi:10.3390/children9071068_

Round 1

Reviewer 1 Report

Dear authors,

Thank you very much for your effort in creating this paper and the questionnaire.

I have no specific remarks to add to your work except for one minor suggestion, please see below

Wish you luck in the further processing of your article

Best regards

Reviewer

Page 5. Line 202.

Please define the abbreviation of OS the first time it’s used.

Author Response

Dear Reviewer 1:

Thank you for your review of our manuscript. We have added the definition of OS according to your comments.

Reviewer 2 Report

This is an interesting observational study that aims "to examine nurses' satisfaction comparing four different pharmacological regimens used for pediatric sedation outside of the operating room." I highlight some comments that should be considered to improve its quality and contribution to the body of knowledge available.

1- Title: it is compelling to the readers, but it is not totally coherent with the study design. The title suggests that a nurse's perception could measure the quality of procedural sedation, and one would expect that a gold standard index for sedation quality would be compared with the nurses' scores. By the way, such a gold standard measure does not exist, as it depends on many objective and subjective variables. So, I suggest that the title is more straightforward to the study aim and design.

2- According to the epidemiological definitions for the type of studies, I understand this is a survey (crossectional design) and not a prospective observational study. Or did the nurses give their opinions soon after each procedural sedation session? 

3- Please define procedural sedation in the Introduction, and also specify what is the aim of procedural sedation in your service in the subheading 2.2. I mean, do you aim at deep sedation for the procedurals and at moderate sedation level for premedication.

4- The Methods should be improved with data about the studied variables and others. Please refer to the STROBE guidelines to favor the methods' reproducibility.

5-  What kind of training do the nurses receive? Please state the nurses' competencies in the Methods.

6- Subheading 2.1: please clarify the operational routine to have written consent forms from the respondents, but have them answer the forms anonymously.

7- In the Results, add reliability data for the ordinal items to reduce the limitation of having a non-validated questionnaire.

8- I understand that nurses' beliefs play a role in their opinions and satisfaction. How could nurses remember each outcome of the sedation encounters to build their perceptions of a drug being better or worse than another? Furthermore, in the last paragraph of subheading 3.1, it is stated that the type of sedative had little influence on their satisfaction with procedural sedation. So, I respectfully think that your key discovery is in this paragraph and not in the numbers that compare the different pharmacological drugs. If you agree, I suggest you amend your manuscript to focus more on adverse events and parental/children unsatisfaction than on the drug type.

9- Considering my previous comment, I think the Conclusion should not highlight the drugs.

Author Response

Reply to Reviewer 2

  • Title: it is compelling to the readers, but it is not totally coherent with the study design. The title suggests that a nurse's perception could measure the quality of procedural sedation, and one would expect that a gold standard index for sedation quality would be compared with the nurses' scores. By the way, such a gold standard measure does not exist, as it depends on many objective and subjective variables. So, I suggest that the title is more straightforward to the study aim and design.

As you suggested, we changed the title in “Nurses' perception of the quality of procedural sedation in children comparing different pharmacological regimens”, which better reflects the aim and the design of our study.

  • According to the epidemiological definitions for the type of studies, I understand this is a survey (crossectional design) and not a prospective observational study. Or did the nurses give their opinions soon after each procedural sedation session? 

We collected the results of the survey at a single point in time, so it is correct to define the study as a cross-sectional study.

  • Please define procedural sedation in the Introduction, and also specify what is the aim of procedural sedation in your service in the subheading 2.2. I mean, do you aim at deep sedation for the procedurals and at moderate sedation level for premedication.

In the introduction, we defined procedural sedation, according to the American College of Emergency Physicians definition.

We specified what our policy of procedural sedation is in our service in the subheading 2.2.

  • The Methods should be improved with data about the studied variables and others. Please refer to the STROBE guidelines to favor the methods' reproducibility.

We followed the STROBE guidelines to write the Methods. All the studied variables are included in the Supplementary material 1.

  • What kind of training do the nurses receive? Please state the nurses' competencies in the Methods.

We add nurses’ competencies and training in the Methods.

  • Subheading 2.1: please clarify the operational routine to have written consent forms from the respondents, but have them answer the forms anonymously.

The survey was completed anonymously so no written consent was needed.

  • In the Results, add reliability data for the ordinal items to reduce the limitation of having a non-validated questionnaire.

In order to increase reliability data, the questionnaire was tested on a convenience sample of 30 nurses, assessing the accuracy of the questions and their applicability on a larger scale. Subsequently, the questions were refined or modified based on the recommendations obtained from the initial participants and the survey was administered among a larger group. Then the survey was administered twice to the nurses over a period of six months to guarantee internal consistency through a test-retest reliability.

  • I understand that nurses' beliefs play a role in their opinions and satisfaction. How could nurses remember each outcome of the sedation encounters to build their perceptions of a drug being better or worse than another? Furthermore, in the last paragraph of subheading 3.1, it is stated that the type of sedative had little influence on their satisfaction with procedural sedation. So, I respectfully think that your key discovery is in this paragraph and not in the numbers that compare the different pharmacological drugs. If you agree, I suggest you amend your manuscript to focus more on adverse events and parental/children unsatisfaction than on the drug type.

Our response to this comment is now included in the main text. We also gave less importance to the comparison between the different pharmacological drugs, even if it is often reported in real life by the staff.

  • Considering my previous comment, I think the Conclusion should not highlight the drugs.

Same as above.

Round 2

Reviewer 2 Report

The authors have satisfactorily amended the manuscript.